# Paradigm Change in First-Line Treatment of Recurrent and/or Metastatic Head and Neck Squamous Cell Carcinoma

**DOI:** 10.3390/cancers13112573

**Published:** 2021-05-24

**Authors:** Edith Borcoman, Gregoire Marret, Christophe Le Tourneau

**Affiliations:** 1Department of Drug Development and Innovation (D3i), Institut Curie, 75005 Paris, France; edith.borcoman@curie.fr (E.B.); gregoire.marret@curie.fr (G.M.); 2INSERM U900 Research Unit, 92210 Saint-Cloud, France; 3Faculty of Medicine, Versailles Saint-Quentin-en-Yvelines University, Paris-Saclay University, 78180 Montigny-Le-Bretonneux, France

**Keywords:** head and neck squamous cell carcinoma, first-line treatment, immunotherapy, PD-L1

## Abstract

**Simple Summary:**

Immunotherapy has made a breakthrough in the treatment of patients with recurrent and/or metastatic head and neck squamous cell carcinoma in the second line setting with anti-PD-1/PD-L1 immune checkpoint inhibitors. Furthermore, the KEYNOTE-048 study has established a new standard of care with immunotherapy in the first line setting, leading to a paradigm change for the treatment of patients with recurrent and/or metastatic head and neck squamous cell carcinoma. No study since 2008 could demonstrate an improvement in survival of these patients until the publication of the KEYNOTE-048 study. Here we will decipher this new paradigm in the first-line line setting and discuss associated challenges.

**Abstract:**

Cetuximab, a monoclonal antibody targeting the epidermal growth factor receptor (EGFR) in combination with platinum-based chemotherapy has been for the decade standard of care for the treatment of head and neck squamous cell carcinomas (HNSCC) patients in the first-line recurrent and/or metastatic setting. The KEYNOTE-048 trial published last year established a new paradigm in this setting with the demonstration that immunotherapy should be given either alone or in combination with chemotherapy. Indeed, pembrolizumab, an antiprogrammed cell death 1 (PD-1) immune checkpoint inhibitor, improved overall survival as compared to the EXTREME regimen in patients expressing PD-L1 in the tumor microenvironment, which represents a large majority of the patient population. In this review, we will decipher this important change of paradigm in the first-line treatment of recurrent and/or metastatic HNSCC, and discuss associated challenges.

## 1. Introduction

Head and neck cancers accounted for approximately 890,000 new cases and 450,000 deaths worldwide in 2018, representing the seventh most frequent cancer worldwide [1]. Squamous cell carcinoma is the most frequent histological subtype of head and neck cancers, beside other rare histological types. Risk factors include typically tobacco and alcohol exposure, but also the exposure to more recently identified high-risk types of human papilloma virus (HPV), with an increasing incidence of HPV-induced oropharyngeal cancer. HPV-induced cancers occur predominantly in younger adults, and are associated with a more favorable prognosis, because of a higher efficacy of chemotherapy and radiotherapy. In addition, patients with HPV-induced cancer are generally more fit than patients with HPV-negative disease, who usually present comorbidities induced by chronic tobacco and alcohol intake [2].

Despite advances in prevention, diagnosis and multimodal treatments, recurrent and/or metastatic head and neck squamous cell carcinomas (HNSCC) develop in approximately 50% of patients with locally advanced HNSCC [3]. Patients with recurrent and/or metastatic HNSCC have a very poor prognosis with a median overall survival (OS) of less than one year [4]. In 2008, the EXTREME phase 3 study was the first study in 30 years to demonstrate an improved disease control and OS in the first-line setting in patients with recurrent and/or metastatic HNSCC, with the addition of cetuximab, a monoclonal antibody targeting the epidermal growth factor receptor (EGFR) to platinum-based chemotherapy versus chemotherapy alone [5]. No significant OS improvement in this particular setting was achieved, until the recently published KEYNOTE-048 trial that established a new standard in this setting with the introduction of immunotherapy, either alone or in combination with chemotherapy [6].

Here we will describe the current standards in the first-line treatment for patients with recurrent and/or metastatic HNSCC, with an extensive presentation of immunotherapy data in this setting. Finally, we discuss future perspectives in this setting.

## 2. Standard Treatments before the Era of Immunotherapy for Patients with Recurrent and/or Metastatic HNSCC

### 2.1. The EXTREME Study in Fisrt-Line Setting

The randomized phase 3 EXTREME study established cetuximab plus platinum-based chemotherapy with 5-FU for up to six cycles followed by weekly cetuximab maintenance therapy until progressive disease (PD) as the standard of care for the first-line treatment of patients with recurrent and/or metastatic HNSCC [5]. The EXTREME regimen was the first regimen in 30 years to significantly improve disease control and OS versus chemotherapy (platinum/5-FU) in this setting, with a median OS of 10.1 months versus 7.4 months [5]. Cetuximab is an immunoglobulin (Ig) G1 isotype monoclonal antibody that binds to the extracellular domain of EGFR, thus blocking downstream signaling activation. Cetuximab has been evaluated in HNSCC because EGFR is overexpressed in the vast majority of HNSCC (up to 90% of patients with HNSCC) [7]. It is noteworthy that patients with aggressive tumor features, relapsing within 6 months following the end of chemoradiation for locally advanced disease were not included in this study.

More recently, Guigay et al. showed that the alternative regimen TPEx (docetaxel, cisplatin, cetuximab), might be an option in the first-line recurrent and/or metastatic setting, with a better safety profile and a reduced number of chemotherapy cycles, despite no significant improvement of OS compared to the standard EXTREME regimen [8].

### 2.2. Second-Line Setting

In the second line setting, no standard of care existed before the era of immunotherapy. It was recommended to primarily address patients for inclusion in clinical trials if they had a good performance status (PS). If patients were not fit for a clinical trial, single agent chemotherapy could be proposed such as methotrexate, docetaxel, paclitaxel, capecitabine, and cetuximab as single agent targeted therapy. These treatments produced a median OS ranging from 3 to 6 months and an overall response rate ranging from 6 to 24% [9,10,11,12,13]. Taxanes in combination with cetuximab might also be an option, although not supported by robust clinical trials [14,15].

The treatment strategy was later on revolutionized with the breakthrough of immunotherapy in the second-line setting.

## 3. Paradigm Change in the Era of Immunotherapy for the Treatment of Patients with Recurrent and/or Metastatic HNSCC

### 3.1. Paradigm Change in the Second-Line Setting

Immunotherapy with immune checkpoint inhibitors has recently made a breakthrough in medical oncology in many different types of tumors. Two anti-PD-1 monoclonal antibodies were recently approved in the second-line setting in recurrent and/or metastatic HNSCC, including pembrolizumab which received FDA approval in August 2016 and nivolumab in November 2016 for the treatment of patients with recurrent and/or metastatic HNSCC who have failed platinum-based therapy [16,17].

In the CheckMate 141 study, 361 patients who were pretreated with platinum were randomized 2:1 to receive either nivolumab (*n* =  240) or standard of care (SOC) chemotherapy monotherapy (*n* = 121) including weekly methotrexate 40 mg/m^2^, docetaxel 30 mg/m^2^, or cetuximab 250 mg/m^2^ (400 mg/m^2^ loading dose first) [16]. Notably in this study, patients who experienced disease progression or recurrence within 6 months after the last dose of platinum-containing chemotherapy administered either as adjuvant therapy for the locoregional setting or in the context of primary or recurrent disease could be included.

OS was significantly longer in the nivolumab arm than in the standard therapy arm (hazard ratio (HR) for death, 0.70; 97.73% confidence interval (CI), 0.51 to 0.96; *p* = 0.01), with a median OS of 7.5 months in the nivolumab group versus 5.1 months in the standard therapy group, regardless of PD-L1 status. Notably, around 10 to 20% of patients experienced durable responses.

Subgroup analysis of patients with PD-L1 tumor cell expression score of ≥1% resulted in a median OS of 8.7 months (HR 0.55; 95% CI: 0.36–0.83), suggesting a better efficacy in patients whose tumor cells express PD-L1. However, nivolumab also improved survival in PD-L1-negative tumors. Nivolumab therefore received the FDA and EMA approval for the treatment of patients with recurrent and/or metastatic HNSCC who have failed platinum-based therapy, regardless of the PD-L1 status.

In the KEYNOTE-040 study, 495 patients were randomized 1:1 to either pembrolizumab (*n* = 247) or SOC monotherapy (*n* = 248), including weekly methotrexate 40 mg/m^2^, docetaxel 75 mg/m^2^ every three weeks, or weekly cetuximab 250 mg/m^2^ (400 mg/m^2^ loading dose first) [17]. In contrast to the Checkmate 141 study, patients with disease progression within 3 months following platinum containing chemotherapy for locoregional disease were excluded.

Median OS in the intention-to-treat population was 8.4 months with pembrolizumab versus 6.9 months with standard of care (HR 0.80, 0.65–0.98; nominal *p* = 0.0161). In the subgroup population of patients with PD-L1 tumor proportion score (TPS) of ≥50%, the median OS was 11.6 months in the pembrolizumab arm versus 6.6 months with SOC monotherapy (HR 0.53; 95% CI: 0.35–0.81, *p* = 0.0014), showing an improvement of OS with higher PD-L1 expression within the tumor microenvironment. No OS improvement was reported in patients with a TPS < 50%.

Following these results, both FDA and EMA approved pembrolizumab as single agent for patients progressing on or after platinum-based chemotherapy whose tumors express PD-L1 with a TPS of ≥50%, whereas nivolumab monotherapy was approved regardless of PD-L1 expression.

Interestingly, neither CheckMate 141 nor KEYNOTE-040 showed an improvement in progression-free survival (PFS) in the immune checkpoint inhibitor arm, but showed a significant better OS, as it has already been observed in other tumor types like non-small-cell lung cancer, for example [18].

Furthermore, results in the subpopulation of patients treated with nivolumab as first-line therapy for patients with recurrent and/or metastatic HNSCC who progressed within 6 months of platinum-based multimodal treatment given as curative intent were further published [19]. This subgroup analysis showed that nivolumab as first-line treatment improved OS versus standard chemotherapy, with a median OS of 7.7 versus 3.3 months (HR 0.56, 95% CI 0.33–0.95), although this analysis represented a small group of patients (*n* = 78). These data suggest a benefit of nivolumab in this particular subset of patients with refractory disease. 

Both CheckMate 141 and KEYNOTE-040 showed a better safety profile with immune checkpoint inhibitors in comparison with standard of care chemotherapy. As previously observed with immune checkpoint inhibitors, some patients achieved durable responses, with an overall response rate of 13% to 18%. In regard of these two studies, immunotherapy was defined as the standard of care for patients with recurrent and/or metastatic HNSCC in the second-line setting, who have failed platinum-based therapy. Nivolumab could also be proposed in first-line setting for patients who recurred rapidly after initial multimodal treatment in the locally advanced setting. Immunotherapies with immune checkpoint inhibitors were thereafter assessed in the first-line setting for patients with recurrent and/or metastatic HNSCC.

### 3.2. Paradigm Change in First-Line Setting

No study since the publication of the EXTREME study showed an OS improvement in the first-line setting for the treatment of patients with recurrent and/or metastatic HNSCC until the recent results of the KEYNOTE-048 study [5,6].

The KEYNOTE-048 study is a randomized, open-label phase 3 study assessing the efficacy of pembrolizumab alone, or in combination with chemotherapy, versus the standard of care EXTREME regimen for the treatment of patients with previously untreated recurrent and/or metastatic HNSCC, in the setting of platinum-sensitive patients.

Patients included had a pathologically confirmed squamous cell carcinoma of the oropharynx, oral cavity, hypopharynx or larynx that was recurrent or metastatic and not curable by a local therapy, with a PS score of 0 or 1, and had a least one tumor lesion measurable per Response Evaluation Criteria in Solid Tumors (RECIST) version 1.1. A tumor sample was required for PD-L1 testing. The p16 expression status as an indicator of HPV infection was required for oropharyngeal cancers (patients with non-oropharyngeal cancers were considered as p16-negative).

Importantly, patients who had progressive disease within 6 months following chemoradiation with cisplatin were excluded from this study, like in the EXTREME trial. Patients included in the study were randomized 1:1:1, and stratified by the percentage of PD-L1-expressing tumor cells (TPS ≥50% versus <50%), p16 status for oropharyngeal cancers, and the PS.

In the pembrolizumab alone group, patients received pembrolizumab 200 mg every 3 weeks until disease progression, intolerable toxicity, patient or physician’s decision, or till a maximum of 35 cycles.

In the chemotherapy group, patients were treated according to the EXTREME regimen, with carboplatin (area under the curve 5 mg/m^2^) or cisplatin (100 mg/m^2^) and 5-fluorouracil (1000 mg/m^2^ per day for 4 consecutive days) every 3 weeks for up to six cycles, in combination with cetuximab (400 mg/m^2^ loading dose, then 250 mg/m^2^ per week) until disease progression, intolerable toxicity, patient or physician’s decision, whichever occurred first.

In the pembrolizumab plus chemotherapy group, patients were treated with pembrolizumab with the same regimen of chemotherapy than in the EXTREME regimen for up to 6 cycles, then pembrolizumab alone until disease progression, intolerable toxicity, patient or physician’s decision, or till a maximum of 35 cycles.

In 2016, more than one year after the KEYNOTE-048 study initiation, an amendment updated the pre-specified PD-L1 biomarker assessment from TPS to the combined positive score (CPS), following preliminary results from the two previous KEYNOTE-012 and KEYNOTE-055 phase I-II studies, which showed that inclusion of inflammatory cells, next to cancer cells, into PD-L1 scoring could improve its predictive value. Further analyses of the KEYNOTE-048 study were planned in the CPS ≥ 20, CPS ≥ 1, and total populations, following a complex statistical plan. The CPS, compared to the TPS that only captured the percentage of PD-L1 expression on stained tumor cells, is a scoring method defined as the total number of tumor cells and immune cells (including lymphocytes and macrophages) stained with PD-L1 divided by the number of all viable tumor cells, then multiplied by 100. Therefore, TPS is a percentage whereas CPS is a value.

Primary endpoints were OS and PFS with an evolving statistical analysis plan testing 14 primary hypotheses. Between 20 April 2015, and 17 January 2017, 882 patients were included in the study. The majority of patients had a tumor expressing PD-L1 with a CPS ≥ 1 (around 85%) and had distant metastases (around 70%). Table 1 resumes efficacy results of the KEYNOTE-048 study. Of note, the study was not designed to compare the two immunotherapy arms with or without addition of chemotherapy. Pembrolizumab alone showed a significant improvement of OS in the CPS ≥ 20 population, with a median OS of 14.9 months (95% CI 11.6−21.5) in the pembrolizumab alone group versus 10.7 months (8.8−12.8) in the EXTREME arm (HR 0.61, 95% CI 0.45−0.83, *p* = 0.0007). In the CPS ≥ 1 population, OS was also improved with pembrolizumab single agent showing a median OS of 12.3 months versus 10.3 months in the EXTREME arm (HR 0.78, 95% CI 0.64−0.96, *p* = 0.0086). In the total population, median OS was 11.6 months in the pembrolizumab alone group versus 10.7 months in the EXTREME arm, but the analysis did not meet the threshold for significant superiority (Table 1).

Furthermore, pembrolizumab plus chemotherapy showed a significant improved OS in the total population, with a median OS of 13.0 months (95% CI 10.9–14.7) in the pembrolizumab plus chemotherapy group versus 10.7 months (9.3–11.7) in the EXTREME arm (HR 0.77, 95% CI 0.63–0.93, *p* = 0.0034). In patients with PD-L1 positive tumors, OS was also improved with pembrolizumab plus chemotherapy versus the EXTREME regimen in both the CPS ≥ 20 and the CPS ≥ 1 populations.

In all three populations, no improvement in the PFS or overall response rate were found, neither with pembrolizumab alone nor pembrolizumab with chemotherapy as compared to the EXTREME arm. Interestingly, the duration of response was much longer in the pembrolizumab single agent arm as compared to the EXTREME arm in all three populations (22.6 months versus 4.5 months in the total population).

The safety profile favored the pembrolizumab-alone arm in comparison with the EXTREME arm (grade 3–5 adverse events: 54.7% versus 83.3% respectively), and was similar for both chemotherapy containing arms.

Finally, first-line therapy with pembrolizumab monotherapy significantly improved OS in PD-L1 CPS ≥ 20 and CPS ≥ 1 populations in comparison with the standard of care EXTREME regimen. First-line therapy with pembrolizumab in combination with chemotherapy significantly improved OS in all three populations (CPS ≥ 20, CPS ≥ 1 and total populations). This study is the first study since a decade to show an OS improvement in comparison with the EXTREME regimen. Some patients will derive prolonged benefit from immunotherapy, with long-lasting responses [20]. However, not all patients will respond to immunotherapy and some patients can experiment rapid progression or so-called hyperprogression [21].

### 3.3. Controversy Raised by the KEYNOTE-048 Study

There is a biological rationale for combining immunotherapy with chemotherapeutic agents, as chemotherapy may induce specific immune responses and stimulate positive immunomodulatory effects [22]. It might also allow for rapid tumor shrinkage, especially in case of patients with high tumor burden and/or rapid tumor progression requiring a rapid tumor shrinkage, or for symptomatic patients. Indeed, an important concern has been described for patients treated with immune checkpoint inhibitors as single agent, referred to as hyperprogression [20,21]. Hyperprogression has been described retrospectively from cumulative published studies assessing immune checkpoint inhibitors, defined as a rapid progression in patients after the start of immunotherapy. This phenomenon was reported in 29% of HNSCC patients, especially in those with a locoregional recurrence in the irradiated field [21]. In case of suspected hyperprogression, with clinical deterioration, immunotherapy should be rapidly stopped, in order to avoid missing the possibility to switch for another treatment like salvage chemotherapy when feasible.

The proportion of patients in the KEYNOTE-048 study who had a progressive disease as best response was indeed greater in the pembrolizumab alone group than in the EXTREME arm (41% versus 12% in the total population), and was associated with a shorter PFS. Whether these results support the hyperprogression phenomenon in the pembrolizumab single-agent arm remains to be demonstrated. However, it is clear that pembrolizumab might be deleterious in a subset of patients who could rather benefit from chemotherapy than pembrolizumab as single-agent treatment.

It is noteworthy that in subgroup analyses of KEYNOTE-048, all HRs favored pembrolizumab single agent versus the EXTREME regimen, except for patients with locoregional recurrent disease only (around 30% of patients included in the study) in the total population and in the CPS ≥ 1 population, suggesting that these patients might not be ideal candidates for treatment with pembrolizumab single agent, maybe because of frequently symptomatic and/or life-threatening disease requiring rapid tumor shrinkage. We acknowledge that these preliminary observations do not allow to draw definitive conclusions.

Conversely, patients in the pembrolizumab plus chemotherapy arm showed similar PFS and objective response rates than in the EXTREME regimen arm, suggesting that chemotherapy might still have an effect in patients who did not respond to pembrolizumab. The study was though not designed to compare the two pembrolizumab arms with or without chemotherapy, yet OS curves look similar, raising the hypothesis that pembrolizumab and chemotherapy are only additive (and not synergistic), and that it might not be very different to give chemotherapy in combination with pembrolizumab or after.

It also seems that the benefit of pembrolizumab single agent is related to the level of tumor PD-L1 expression [23]. The benefit of pembrolizumab on survival in patients with a tumor PD-L1 expression with a CPS between 1 and 19 will also need to be further clarified [23]. The question is still pending for patients without PD-L1 tumor assessment in clinical routine. Following the results of the KEYNOTE-048 study, the FDA approved pembrolizumab in combination with chemotherapy as first-line treatment regardless of PD-L1 expression and pembrolizumab single agent for patients with PD-L1-expressing tumors (CPS ≥ 1). Conversely, the EMA approved pembrolizumab with or without chemotherapy only for patients with a CPS ≥ 1, based on the fact that neither pembrolizumab single agent nor pembrolizumab with chemotherapy showed an OS benefit in the CPS = 0 population, as shown in a post hoc analysis [24]. In the CPS ≥ 1 to CPS < 20 population (representing 40% of the population) pembrolizumab plus chemotherapy showed a significant advantage in OS (12.7 vs. 9.9 months, HR, 0.71; 95% CI, 0.54–0.94), but pembrolizumab alone did not show a significant benefit (10.8 vs. 10.1 months, HR, 0.86; 95% CI, 0.66–1.12). However this post hoc analysis might be interpreted with caution, since it was not preplanned. These different points suggest the need for a careful selection of patients who might be eligible for pembrolizumab single agent treatment, to avoid a deleterious choice that might be harmful.

### 3.4. Practical Recommandation for Treatment Algorithm

Results from the KEYNOTE-048 have recently been incorporated in the ESMO clinical practice guidelines for the treatment of recurrent and/or metastatic HNSCC patients [25]. Both Figure 1 and Figure 2 resume a proposed algorithm in the fist-line and second-line settings respectively.

Besides platinum-sensitivity/resistance and tumor PD-L1 status, treatment decision need to take into account PS and patient’s preferences. Furthermore, it seems important to take into account the kinetic of disease progression, and whether or not the patient presents with symptomatic and/or life-threatening tumor localization, as it might require rapid tumor shrinkage and may lead clinicians to prefer the combination of pembrolizumab with chemotherapy, if patient is fit for platinum-based chemotherapy. Notably, median time to response was similar between nivolumab arm and chemotherapy arm in the CheckMate 141 study [16]. Data regarding time to response and time to deterioration will need to be clarified from the KEYNOTE-048 study to help clinicians in treatment choice.

No recommendations exist on how to treat recurrent and/or metastatic HNSCC patients after exposure to first-line immunotherapy +/− chemotherapy, as no published data exist for now. Figure 3 summarizesa proposed treatment algorithm after exposure to first-line immunotherapy. 

A retrospective study of 82 French patients showed an overall response rate of 30%, a median PFS of 3.6 months and a median OS of 7.8 months with salvage chemotherapy for patients who have failed immune checkpoint inhibitors [15]. These data suggest that immunotherapy may increase sensitivity to subsequent lines of treatment, since survival and tumor responses seem better than historical data without first-line immunotherapy [15].

These results also imply that every patient with a diagnosis of HNSCC will need to have a tumor PD-L1 assessment using the CPS score (and not the TPS) for future treatment decisions, as this was not done in clinical routine. Further challenges will need to be overcome regarding the reliability of PD-L1 assessment, like tumor heterogeneity, the choice of the best timing to assess PD-L1 status (at initial diagnosis of the primary tumor versus at diagnosis of recurrence), the standardization of assay and cut-off used [26].

## 4. Future Perspectives

### 4.1. Other Immune Checkpoint Inhibitors

Because only a minority of patients will respond to immune checkpoint inhibitors, there is an urgent need to improve antitumor immune response. With this goal to improve the antitumoral efficacy of anti-PD-1 immune checkpoint inhibitors for the treatment of patients with recurrent and/or metastatic HNSCC after progression to platinum-based regimen, several new immune checkpoint inhibitors have been investigated alone or in association with anti-PD-1 agents with disappointing results. This is the case for example for the lirilumab, a killer-cell immunoglobulin-like receptors (KIRs) inhibitor, blocking the interaction between inhibitor receptors KIR2DL1, KIR2DL2, KIR2DL3 with their ligands, allowing to stimulate the activation of natural killer lymphocytes, assessed in phase II randomized study showing negative results (NCT01714739). Another immune checkpoint inhibitor, epacadostat, an indoleamine 2,3-dioxygenase-1 (IDO1) inhibitor (KEYNOTE-037), has been assessed in association with pembrolizumab, but the development was stopped after the negative results of a randomized phase III study published in patients with advanced melanoma [27]. The EAGLE study, phase III assessing the association of durvalumab (anti PD-L1) with tremelimumab (anti-CTLA-4) versus treatment at the investigator‘s choice (cetuximab, a taxane, methotrexate, or a fluoropyrimidine) in patients pre-treated with platinum based chemotherapy also showed negative results [28]. The same combination of durvalumab plus tremelimumab, or durvalumab monotherapy were further assessed in the first line setting for patients with recurrent or metastatic HNSCC not previously pretreated in the phase III KESTREL study, versus standard EXTREME chemotherapy regimen (NCT02551159), with early communicated results suggesting lack of OS benefit in the immunotherapy arms.

Other immunotherapies have proved interesting preliminary results, like GSK609, an agonist antibody aiming to stimulate the anti-tumoral immune response by targeting the inducible T cell co-stimulatory receptor (ICOS), and are summarized in Table 2 [29]. The ICOS agonist is currently evaluated in a phase III study in combination with pembrolizumab versus pembrolizumab plus placebo in patients with CPS ≥ 1 recurrent and/or metastatic HNSCC (INDUCE-3 study) in the first line setting, or in combination with chemotherapy platinum/5-FU + pembrolizumab (INDUCE-4 study) [29,30].

Another interesting molecule is the monalizumab, a first-in-class immune checkpoint inhibitor targeting NKG2A receptors expressed on tumor infiltrating cytotoxic CD8+ T cells and NK cells, aiming to unleash NK an effector CD8+ T cells antitumoral immune response, currently assessed in combination with cetuximab in recurrent and/or metastatic HNSCC patients previously treated with platinum-based chemotherapy and PD-(L)1 inhibitors [31]. In the phase II expansion cohort, results reported by Cohen RB et al. at ASCO 2020 showed a promising activity with response rate of 20%, and 6-month and 12-month OS rate of 80% and 33% respectively with a manageable safety profile, leading to a phase III study to confirm these results in heavily pretreated patients already exposed to platinum and immune checkpoints inhibitor agents.

Another strategy aiming to improve efficacy of immunotherapy agents, beside the combination of two immunotherapy agents, or combination with chemotherapy or targeted therapies already approved in the treatment of HNSCC, is to assess the effect of an epidrug, like vorinostat, a histone DesACetylases (HDAC) inhibitor in combination with immune checkpoint inhibitors, as preclinical evidence has suggested that modulating the epigenome might improve the efficacy of immunotherapy (NCT04357873) [32].

### 4.2. Immunotherapeutic Vaccines

Immunotherapeutic vaccines are currently developed in the population of recurrent and/or metastatic HPV+ HNSCC with promising preliminary results. In this particular population, the aim is to prime the immune system after the vaccine injection to stimulate a specific T cells immune response against HPV oncoproteins E6 and E7, with the aim to improve antitumoral immune response. Several technologies are investigated with the development of different vaccine molecules, like Tipapkinogene sovacivec (TG4001), a vaccine using an attenuated and modified poxvirus (MVA) as a vector expressing the HPV16 E6 and E7 proteins (rendered nononcogenic) and interleukin (IL)-2 assessed in a phase I/II trial in combination with avelumab in patients with HPV-16 positive refractory solid tumors (NCT03260023) [33], or MEDI0457, a DNA immunotherapeutic vaccine expressing HPV 16/18 E6/E7 proteins and IL-12 assessed in combination with durvalumab in a phase I/II trial in patients with HPV-positive recurrent and/or metastatic HNSCC, who have received one or more prior platinum-containing regimen [34]. The induced co-expression of an IL molecule will boost the immune response and stimulate the production of cytotoxic T cells to increase tumor cell death.

The future development of treatments for patients with recurrent and/or metastatic HNSCC will certainly give access to new modality of treatments with the goal to improve antitumor immune response of already approved immune checkpoint inhibitors, either with new immune checkpoint molecules or new combinations.

### 4.3. Immunotherapy in the Early Phase Setting

Another axis of development for immunotherapy molecules is to bring these agents to the early phase setting, for the treatment of patients with locally advanced HNSCC, in combination with radiotherapy. The JAVELIN phase III study assessed the combination of avelumab with concurrent chemoradiotherapy followed by avelumab maintenance in this setting [35]. At the interim analysis both PFS and OS hazard ratios were in favor of the placebo arm, with higher grade ≥3 adverse events in the avelumab arm, showing no benefit of adding an anti-PD-L1 antibody to standard treatment. In a phase II study, pembrolizumab was assessed in association with radiotherapy versus cetuximab plus radiotherapy in patients with locally advanced HNSCC in patients unfit for cisplatin chemotherapy [36]. In this study, pembrolizumab did not improve loco-regional control rate neither survival, but appeared less toxic (74% vs. 92% patients with at least one grade ≥3 acute adverse event).

Further studies are ongoing in the locally advanced setting, for example assessing avelumab in combination with cetuximab and radiotherapy (NCT02999087,) so far no changing in standard of care treatments has been achieved with immunotherapy agents.

Immunotherapies are currently also assessed in the neodjuvant setting, showing promising response rates, without any surgical delays, with nivolumab or nivolumab plus ipilimumab in the neoadjuvant setting for the treatment of patients with resectable squamous cell carcinoma of the oral cavity (≥T2, or clinically node positive), or with pembrolizumab for the treatment of resectable HPV-negative, Stage III/IV HNSCC [37,38]. The KEYNOTE 689 randomized phase III study is currently evaluating pembrolizumab as neoadjuvant and adjuvant therapy in combination with standard of care chemoradiotheray in patients with previously untreated, resectable locally advanced HNSCC (NCT03765918). Whether introducing an immunotherapy agent before surgery and radiotherapy, to potentially stimulate a better antitumor immune response before removing with surgery or killing immune cells in the tumor microenvironment with radiotherapy will translate in a significant clinical benefit is still not clear, and these results are particularly awaited.

## 5. Conclusions

The phase 3 KEYNOTE-048 has been the first study to change the standard of care for the treatment of patients with recurrent and/or metastatic HNSCC in the first-line setting since 2008 after the publication of the EXTREME study. In the KEYNOTE-048 study, pembrolizumab combined with chemotherapy (5FU + cisplatin or carboplatin) was associated with an increased OS and a comparable safety profile than the EXTREME regimen. Furthermore, pembrolizumab single agent showed a prolonged OS in patients with PD-L1 positive tumors (CPS ≥ 1) and a better safety profile. The likelihood to respond to pembrolizumab single agent is lower than to the EXTREME regimen, however, responses are often durable with immunotherapy. The choice between one or the other regimen, adding or not chemotherapy to pembrolizumab, must be made carefully by physicians taking into account PS and patient’s choice, beside PD-L1 tumor status and platinum-sensitivity. Patients with locoregional recurrence only, with symptomatic disease, rapid progression or life-threatening tumor localization might be treated preferentially with the combination of pembrolizumab and chemotherapy rather than pembrolizumab alone to avoid the risk of hyperprogression, if they are eligible for platinum-based treatment. KEYNOTE-048 study also modifies the treatment algorithm for subsequent lines of therapy; although it might seem logical to propose the EXTREME regimen in patients who have failed pembrolizumab single agent in first-line. Further long-term data will help clinicians to fine-tune decisions in the treatment algorithm. Moreover, several new immunotherapy agents and new combinations are currently being evaluated in the treatment of patients with advanced HNSCC, with the goal to improve antitumoral immune response and increase the rate of long responders to immune checkpoint inhibitors, with several of these approaches already showing promising results, that might in the future modify treatment algorithms and improve outcomes.

## Figures and Tables

**Figure 1 cancers-13-02573-f001:**
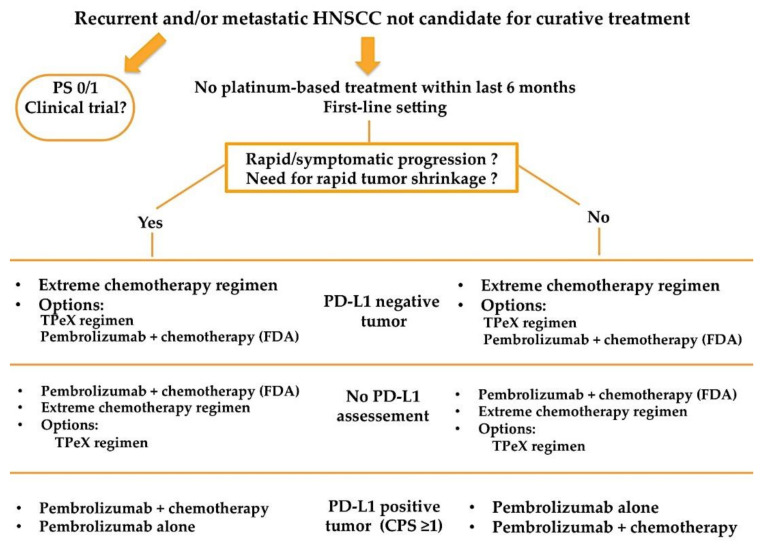
Proposed treatment algorithm in the first line setting for patients with recurrent and/or metastatic HNSCC.

**Figure 2 cancers-13-02573-f002:**
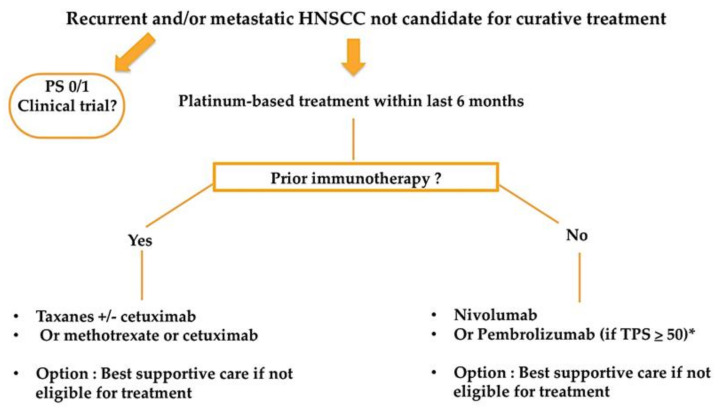
Proposed treatment algorithm in the second-line setting for patients with recurrent and/or metastatic HNSCC. * Patients with progression within 3 months following platinum containing chemotherapy for locoregional disease excluded.

**Figure 3 cancers-13-02573-f003:**
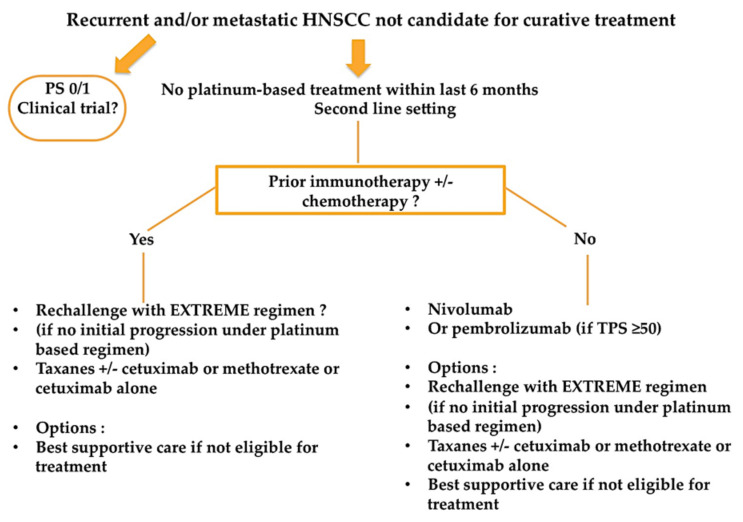
Proposed treatment algorithm for recurrent and/or metastatic HNSCC patients after exposure to first-line immunotherapy +/− chemotherapy.

**Table 1 cancers-13-02573-t001:** Resumed efficacy results of the KEYNOTE-048 phase 3 study.

Variations	Treatment Arm		
	Pembrolizumab Alone *n* = 301	EXTREME Regimen *n* = 300 *	Pembrolizumab + Chemotherapy *n* = 281
	All/CPS ≥ 1/CPS ≥ 20	All/CPS ≥ 1/CPS ≥ 20	All/CPS ≥ 1/CPS ≥ 20
Median OS (months)	11.6/12.3 ^+^/14.9 ^+^ vs. 10.7/10.3/10.7		
		10.7/10.4/11.0 vs. 13.0 ^+^/13.6 ^+^/14.7 ^+^	
Median PFS (months)	2.3/3.2/3.4 vs. 5.2/5.0/5.0		
		5.1/5.0/5.2 vs. 4.9/5.0/5.8	
Overall Response Rate (%)	17/19/23 vs. 36/35/36		
		36/36/38 vs. 36/36/43	
Median DOR (months)	22.6/23.4/22.6 vs. 4.5/4.5/4.2		
		4.3/4.3/4.2 vs. 6.7/6.7/7.1	

All = all population; VS = versus; CPS = combined positive score; OS = overall survival; PFS = progression-free survival; DOR = duration of response; + = statistical significant improvement. ***** Of note, 300 patients were assigned to the EXTREME arm, 300 patients were included in the intention-to-treat population for comparison with the pembrolizumab alone arm, and 278 among them were included in the intention-to-treat population for comparison with the pembrolizumab + chemotherapy arm, explaining the different results obtained in the control arm for each comparison.

**Table 2 cancers-13-02573-t002:** Future perspectives of immunotherapies in the treatment of HNSCC patients.

Study	Clinical Setting	Study Drug (s)	Results
Anti PD-1/PD-L1 Inhibitors Combinations			
Phase II (PEVOsq)	Patient with recurrent and/or metastatic squamous cell carcinoma of the head and neck, lung, cervix, anus, vulva, and penis	Pembrolizumab + vorinostat (histone deacetylases (HDAC) inhibitor)	Ongoing
Phase II (PembroRAD)	Patients with nonoperated stage III-IVa locally advanced HNSCC and unfit for receiving high dose cisplatin	Pembrolizumab in combination with radiotherapy vs. cetuximab in combination with radiotherapy	LRC at 15 months after radiotherapy: 60% vs. 59%, OR = 1.05, *p* 0.91. No significant difference in PFS and OS (36)
Phase III (JAVELIN HN100)	Patients with nonoperated stage III-IVa or IVb locally advanced HNSCC	Avelumab in combination with concurrent chemoradiotherapy followed by avelumab maintenance vs. placebo with chemoradiotherapy followed by placebo maintenance	HR for PFS 1.21 (95% CI: 0.93–1.57; *p* = 0.92) in favor of placebo arm (35)
Phase II	Patients with untreated squamous cell carcinoma of the oral cavity (≥T2, or clinically node positive) in the neoadjuvant setting	Nivolumab weeks 1 and 3 vs. nivolumab + ipilimumab, surgery 3 to 7 days after cycle 2	RECIST response 13% vs. 38%. Pathologic response 54% vs. 73% in favor of the combo (37)
Phase II	Patients with resectable HPV-negative stage III/IV HNSCC	Pembrolizumab one cycle neoadjuvant. Only for patients with high-risk pathologic features: pembrolizumab following standard adjuvant chemoradiotherapy	Pathologic response 43% (38)
Phase III (KEYNOTE 689)	Patients with locally advanced resectable HNSCC	Pembrolizumab one cycle neoadjuvant followed by surgical resection then SOC plus adjuvant pembrolizumab (15 cycles) vs. surgical resection followed by adjuvant SOC	Ongoing
**Other immune checkpoint inhibitors**			
Phase II Phase III ongoing	HNSCC patients who have progressed after platinum-based chemotherapy and anti-PD-1 inhibitors	Monalizumab (NKG2A inhibitor) + cetuximab vs. placebo + cetuximab	ORR 20%. 8 PR/40. Median TTR 1.6 months (31)
Phase I expansion cohort (INDUCE-1)	HNSCC patients who have progressed after platinum-based chemotherapy, ≤5 prior lines of therapy for advanced disease	GSK609 (ICOS agonist) for anti PD-1/PD-L1 experienced patients and + pembrolizumab for anti PD-1/PD-L1 naïve patients	- Monotherapy: ORR 8% (1/8). - Combination: ORR 28% (8/29). Median PFS: 5.6 months (29)
Phase III (INDUCE-3)	Patients with PD-L1 CPS ≥ 1 recurrent and/or metastatic HNSCC in the first line setting	GSK609 (ICOS agonist) + pembrolizumab vs. placebo + pembrolizumab	On hold
Phase III (INDUCE-4)	Patients with recurrent and/or metastatic HNSCC in the first line setting	GSK609 (ICOS agonist) + chemotherapy platinum/5-FU + pembrolizumab vs. placebo + chemotherapy platinum/5-FU + pembrolizumab	On hold
**Immunotherapeutic vaccines**			
Phase I/II	Patients with HPV-16/18 positive HNSCC who have progressed after platinum-based chemotherapy	MEDI0457 vaccine + durvalumab	ORR 22%, PR 3/27, CR 3/27, SD 6/27, PD 13/27 (34)
Phase I/II (TG4001.12)	Patients with refractory HPV-16 positive refractory solid tumors	Tipapkinogene sovacivec (TG4001) HPV16 vaccine + avelumab	ORR 23% (33)

ORR = overall response rate; PR = partial response; TTR = time to response; PFS = progression free survival; CR = complete response; SD = stable disease; PD = progressive disease; SOC = Standard of care.

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
