# Peer review of "Paradigm Change in First-Line Treatment of Recurrent and/or Metastatic Head and Neck Squamous Cell Carcinoma"

_cancers, 2021, doi:10.3390/cancers13112573_

Round 1

Reviewer 1 Report

The authors have addressed the comments. Specifically, the addition of combination checkpoint inhibitors and other immunotherapies strategies have enhanced the breath of the paper beyond Keynote-48 and Checkmate 141 studies.

A minor comment is to table the results of these combination treatments; and that of Neo-adjuvant immunotherapies in locally advanced HNSCC. Otherwise I have no further comments

Author Response

We thank reviewer 1 for his/her feedback.

We have added a table that resumes new IO strategies, referred as Table 2 in the manuscript.

Reviewer 2 Report

The authors have appropriately responded to prior criticisms.

Author Response

We thank reviewer 1 for his/her feedback.

Reviewer 3 Report

The paper is suitable for publication.

Author Response

We thank reviewer 3 for his/her feedback.

Reviewer 4 Report

I really appreciate the work of Borcoman E and coworkers who aimed to summarize and simplify the treatment options of recurrent or metastatic HNSCCs. This is definitely justified since treatment has changed significantly within the past years. 

First of all, I would really recommend to go through the work another time to revise typos (there are numerous) and wording, which makes reading more exhausting than required. 

Typos (few examples)

Figure Legend 1: algorithm in the fist-line -> in the first

Line 343: overal instead of overall 

Line 349: curently instead of currently

Line 369: several news immune instead of several new immune

Line 295: alone dit not instead of did not

Wording: 

Line 443:  (...) but for now - currently, until now, so far sounds better

Line 348: (...) that every patient - each patient 

Legend Figure 3 (...) chemotherapy for locoregional disease were excluded

However, these are of course just minor comments. Some more important comments are indicated as follows: 

1) What is the difference between TPS and CPS? Could the authors please provide a section dealing with this topic please include which score has to be applied in which particular scenario? 

2) What is the definition of first and second line? For non-oncologists this would be of interest.

3) Regarding Table 1 - as stated patients were randomized 1:1:1 to the Pembro single arm, the EXTREME arm and the Pembro + ChT arm. Why do the values of the EXTREME group differ between analysis to Pembro alone or Pembro + ChT arm? I may be wrong, but I suppose that these have been the same patients? 

4) Sometimes comprehensive tables would be more appropriate than listing all results of studies: e.g. Lines 224 to 232

5) Figure 1-3: please verify the abbreviation PS 0/1

6) Figure 1: Use bold letters for EXTREME as done during the manuscript

7) Chapter 4 - is really interesting, but they authors do provide just too much data impairing eligibility. Again, less data or smart tables would be great. Maybe it would be better to create a table listing the most promising candidates....

8) Conclusion is much too long - is should not be a second discussion. For clinicians it would be of interest to know, wich patient should be treated with which agent and what are potential drawbacks or controversies. For Instance Pembro alone - response rate, durability vs. hyperprogression and why to not apply the Pembro + ChT schema - toxicity, but safer and so on. This would be most interesting for clinical routine. 

Altogether, I really believe that the current work is worthwhile to be considered for publication, but extensive english editing and revisions are necessary. It is crucial to shorten some sections, while discussing others more extensively.  

I really looking forward to receive a revised version for reviewing. 

Author Response

Reviewer 4

I really appreciate the work of Borcoman E and coworkers who aimed to summarize and simplify the treatment options of recurrent or metastatic HNSCCs. This is definitely justified since treatment has changed significantly within the past years.

First of all, I would really recommend to go through the work another time to revise typos (there are numerous) and wording, which makes reading more exhausting than required.

Typos (few examples)

Figure Legend 1: algorithm in the fist-line -> in the first

Line 343: overal instead of overall

Line 349: curently instead of currently

Line 369: several news immune instead of several new immune

Line 295: alone dit not instead of did not

Wording:

Line 443:  (...) but for now - currently, until now, so far sounds better

Line 348: (...) that every patient - each patient

Legend Figure 3 (...) chemotherapy for locoregional disease were excluded

We thank the Reviewer for his/her useful feedback.

We have reviewed the manuscript and corrected all typos and English wording.

However, these are of course just minor comments. Some more important comments are indicated as follows:

1) What is the difference between TPS and CPS? Could the authors please provide a section dealing with this topic please include which score has to be applied in which particular scenario?

We have added a paragraph defining TPS and CPS as follow line 205:

The CPS, compared to the TPS that only captured the percentage of PD-L1 expression on stained tumor cells, is a scoring method defined as the total number of tumor cells and immune cells (including lymphocytes and macrophages) stained with PD-L1 divided by the number of all viable tumor cells, then multiplied by 100. Therefore, TPS is a percentage whereas CPS is a value.

And we modified the text as follow line 326:

These results also imply that every patient with a diagnosis of HNSCC will need to have a tumor PD-L1 assessment using the CPS score (and not the TPS) for future treatment decisions.

2) What is the definition of first and second line? For non-oncologists this would be of interest.

We thanks reviewer for his/her comment but we think that defining these concepts of first and second line is not necessary for this manuscript.

3) Regarding Table 1 - as stated patients were randomized 1:1:1 to the Pembro single arm, the EXTREME arm and the Pembro + ChT arm. Why do the values of the EXTREME group differ between analysis to Pembro alone or Pembro + ChT arm? I may be wrong, but I suppose that these have been the same patients?

Regarding Table 1 we have added some precisions in the legend of Table 1 to clarify this question.

* Of note, 300 patients were assigned to the EXTREME arm, 300 patients were included in the intention-to-treat population for comparison with the pembrolizumab alone arm, and 278 among them were included in the intention-to-treat population for comparison with the pembrolizumab + chemotherapy arm, explaining the different results obtained in the control arm for each comparison.

4) Sometimes comprehensive tables would be more appropriate than listing all results of studies: e.g. Lines 224 to 232

We have used Tables to summarize the results like in Table 1, and added in the manuscript another table as requested by the first Reviewer (Table 2).

5) Figure 1-3: please verify the abbreviation PS 0/1

The definition of PS is in the text line 94.

6) Figure 1: Use bold letters for EXTREME as done during the manuscript

We have modified the Figure 1 as suggested.

7) Chapter 4 - is really interesting, but they authors do provide just too much data impairing eligibility. Again, less data or smart tables would be great. Maybe it would be better to create a table listing the most promising candidates....

We have added a Table to resume the most promising future perspectives, referred as Table 2 in the manuscript.

8) Conclusion is much too long - is should not be a second discussion. For clinicians it would be of interest to know, wich patient should be treated with which agent and what are potential drawbacks or controversies. For Instance Pembro alone - response rate, durability vs. hyperprogression and why to not apply the Pembro + ChT schema - toxicity, but safer and so on. This would be most interesting for clinical routine.

Altogether, I really believe that the current work is worthwhile to be considered for publication, but extensive english editing and revisions are necessary. It is crucial to shorten some sections, while discussing others more extensively. 

We thank the reviewer for his/her useful feedback. The whole text has been remodeled for English editing, and we have shortened some parts, including the conclusion, to make the manuscript easier to read.

The conclusion has been modified as follows:

The phase 3 KEYNOTE-048 has been the first study to change the standard of care for the treatment of patients with recurrent and/or metastatic HNSCC in the first-line setting since 2008 after the publication of the EXTREME study. In the KEYNOTE-048 study, pembrolizumab combined with chemotherapy (5FU + cisplatin or carboplatin) was associated with an increased OS and a comparable safety profile than the EXTREME regimen. Furthermore, pembrolizumab single agent showed a prolonged OS in patients with PD-L1 positive tumors (CPS ≥1) and a better safety profile. The likelihood to respond to pembrolizumab single agent is lower than to the EXTREME regimen, however, responses are often durable with immunotherapy. The choice between one or the other regimen, adding or not chemotherapy to pembrolizumab, must be made carefully by physicians taking into account PS and patient’s choice, beside PD-L1 tumor status and platinum-sensitivity. Patients with a locoregional recurrence only, symptomatic disease, rapid progression or life-threatening tumor localization might be treated preferentially with the combination of pembrolizumab and chemotherapy rather than pembrolizumab alone to avoid the risk of hyperprogression, if they are eligible to platinum based treatment. KEYNOTE-048 study also modifies the treatment algorithm for subsequent lines of therapy; although it might seem logical to propose the EXREME regimen in patients who have failed pembrolizumab single agent in first-line. Further long-term data will help clinicians to fine-tune decisions in the treatment algorithm. Moreover, several new immunotherapy agents and new combinations are currently being evaluated in the treatment of patients with advanced HNSCC, with the goal to improve anti-tumoral immune response and increase the rate of long responders to immune checkpoint inhibitors, with several of these approaches already showing promising results, that might in the future modify treatment algorithms and improve outcomes.

Round 2

Reviewer 4 Report

Thanks to the authors who have addressed all raised points adequately. Now, I think that the work is suitable for publication. 

This manuscript is a resubmission of an earlier submission. The following is a list of the peer review reports and author responses from that submission.

Round 1

Reviewer 1 Report

Thank you for the opportunity to review this work. This review encompasses a nice overview of the recent clinical trials and summarises the results of Checkmate-141; KEYNOTE 040 and KEYNOTE 048 in the treatment of patients with recurrent/metastatic HNSCC (RM-HNSCC), either as the first or second line treatment options. Incorporation of these landmark studies in the paradigm of clinical care was presented. 

The summary was concise and detailed, and highlighted some subtleties in analyses of these studies. Table 1 was a good summary of the KEYNOTE 048 study but Figures 1 to 3 could potentially be condensed into a single figure. Nevertheless, this paper did not add further insights towards the current understanding of managing patients with RM-HNSCC. Notably, a recent paper published in Annals Oncology by EHNS-ESMO-ESTRO working group had done a similar summary on this topic (see Figure 5) (JP Machiels et al Ann Onco Nov 2020). 

Without further insights from the authors (especially on looking ahead of possible challenges and incorporating immunotherapy( beyond PD1 mAb) into the current paradigm), it is difficult to publish this work in its current form. 

Reviewer 2 Report

This is a well-written, succinct but comprehensive, review of first and second line therapies for recurrent/metastatic HNSCC.  The supplanting of cetuximab-based by immune checkpoint-based approaches is described, along with the supporting clinical trial rationale for the changes.  Potential points of concern/consideration (eg. hyperprogression) are described.  The review will be particularly helpful for clinicians looking for the foundational rationale for current standard of care incorporating immune checkpoint inhibitors.

Only minor typographical changes are needed:

1) line 61:  correct:  "frist"

2)  line 116:  what is meant by "did also better"?

3) line 201:  what is meant by "Table 1 resume efficacy"?  A similar issue on line 301 with the word "resume".

4)  line 211:  correct:  "threeshold"

5)  line:  301:  correct:  "frist"

6)  line 303:  correct: "perfomance"

7)  line 339:  correct:  "cutt"

Reviewer 3 Report

Paradigm change in first-line treatment of recurrent and/or metastatic head and neck squamous cell carcinoma

This review article focuses on the treatment of recurrent and/or metastatic HNSCC.

The authors, in order to analyze the issues linked to the recently published KEYNOTE-048 trial, provided an interesting excursus about the possible therapeutic options, both in a first and in a second-line setting, in case of malignant recurrency or metastasis.

The authors carefully analyzed the results of the above-mentioned clinical trial, without neglecting the controversies that followed this “paradigm change”.

The final section of the review provides the proposal of three treatment algorithms, according to the different clinical conditions of patients, to treat HNSCC recurrency or metastasis by the administration of pembrolizumab, but few evidences are available.

The article is well written and could be of some interest especially if widened with a systemic review strategy, but it is not of prior interest for this Journal and seems more like an expert opinion than a scientific article.